SOFTWARE

# EEG-Pype: An accessible MNE-Python pipeline with graphical user interface for preprocessing and analysis of resting-state electroencephalography data

D. Yorben Lodema[1]*, Herman J. van Dellen[1], Willem de Haan[2], Margot van Hest[1], Arjan Hillebrand[3,4,5], Edwin van Dellen[1,6]

**1** Department of Psychiatry, University Medical Center Utrecht, Utrecht, the Netherlands, **2** Alzheimer Center and Department of Neurology, Amsterdam Neuroscience, VU University Medical Center, Amsterdam UMC, Amsterdam, the Netherlands, **3** Amsterdam Neuroscience, Brain Imaging, Amsterdam, the Netherlands, **4** Amsterdam Neuroscience, Systems and Network Neurosciences, Amsterdam, the Netherlands, **5** Department of Clinical Neurophysiology and MEG Center, Department of Neurology, Amsterdam Neuroscience, Vrije Universiteit, Amsterdam, the Netherlands, **6** Department of Neurology, UZ Brussel and Vrije Universiteit Brussel, Brussels, Belgium

* y.lodema@outlook.com

## Abstract

Processing of electroencephalography (EEG) data requires multiple steps to remove noise and artifacts and select good-quality data. While powerful open-source tool-boxes like MNE-Python exist, their command-line nature can pose a barrier for researchers without programming experience. Here, we present EEG-Pype, an open-source (Apache-2.0 licensed) graphical user interface application using MNE-Python functions. EEG-Pype provides an intuitive workflow tailored for preprocessing of resting-state EEG data, including frequency band filtering, independent component analysis and atlas-based beamforming for source-level analysis. The application supports several common raw EEG input file formats and guides users through a comprehensive pipeline focused on manual bad channel and epoch selection. Manual steps are streamlined using MNE-Python's interactive plots, resulting in a user-friendly experience. Configuration saving and loading allows for batch (re)runs, while a separate log is also saved, improving reproducibility and documentation. Output can be saved after filtering in canonical frequency bands, ready for further analysis. EEG-Pype includes a module for calculating quantitative EEG measures on preprocessed data, including spectral, functional connectivity and network analysis metrics. This software aims to lower the entry barrier for standardized EEG preprocessing, promoting reproducible research practices among neuroscientists and clinicians without requiring programming knowledge. EEG-Pype can be downloaded from: https://github.com/yorbenlodema/EEG-Pype and is not dependent on a specific operating system.

of the Creative Commons Attribution License, which permits unrestricted use, distribution, and reproduction in any medium, provided the original author and source are credited.

**Data availability statement:** All relevant data are within the manuscript and its Supporting Information files, excluding our test.bdf EEG file due to file size constraints. This file is, and will stay, available on our GitHub page: [https://github.com/yorbenlodema/EEG-Pype](https://github.com/yorbenlodema/EEG-Pype).

**Funding:** The author(s) received no specific funding for this work.

**Competing interests:** The authors have declared that no competing interests exist.

## Author summary

We developed EEG-Pype, a free and open-source software tool, to make the complex analysis of brain electrical activity (electroencephalography, or EEG) more accessible to the broader scientific community. When EEG is recorded, the raw data is inevitably contaminated by artifacts from sources like muscle activity or eye movement. The essential steps to clean the data before analysis, often called preprocessing, typically require significant programming expertise. This technical barrier can prevent researchers from directly processing their own data, creating a bottleneck in analysis. Our software addresses this challenge by offering an intuitive graphical user interface that provides a guided, end-to-end workflow for EEG preprocessing and analysis. It uses powerful, established computational libraries for filtering, artifact removal, and source analysis, all without requiring manual coding. We focused on resting-state EEG, a key modality for studying intrinsic brain networks. EEG-Pype also includes a module to compute advanced quantitative metrics, such as functional connectivity and network topology, directly from the processed data. By lowering the technical barrier to sophisticated EEG analysis, EEG-Pype facilitates more transparent, standardized, and reproducible neuroscience.

## Introduction

Electroencephalography (EEG) is a widely used neurophysiological technique in neuroscience and clinical practice, offering millisecond resolution measurements of brain dynamics. Raw EEG data is contaminated by artifacts such as muscle activity, eye blinks, cardiac artifacts and line noise, and requires careful preprocessing to isolate neural signals of interest. EEG preprocessing requires significant knowledge about signal analysis, the nature of EEG artifacts and often programming experience.

While powerful toolboxes for EEG (pre)processing exist, they present different barriers to entry. Tools like EEGLAB [1] and Brainstorm [2] can make EEG preprocessing more accessible but are built in MATLAB, a proprietary software not freely accessible. MNE-Python [3] is an open-source Python package that is free to use and offers a wide array of functions for powerful EEG and magnetoencephalography (MEG) analyses. However, analysis is primarily achieved through command-line instructions and construction of Python scripts. This requires programming experience, which may provide a significant hurdle for many researchers, students, and clinicians. Tools like DIGEEG and later BrainWave [4], developed by C.J. Stam, fit in a tradition of user-friendly EEG analysis tools for clinical researchers. These tools utilized a workflow centered on visual inspection for preprocessing and quantitative EEG (qEEG) analyses. However, being built on pre-compiled Java code, largely dependent on a single developer, this ecosystem faces challenges concerning open-source development and modifiability. There is therefore a need for a successor that embodies this spirit of accessibility but is built on a modern, open-source, and community-driven platform.

With these considerations in mind, a standardized workflow and accompanying user-friendly software can stream-line preprocessing to make visual resting-state EEG inspection easier to apply. Therefore, we developed EEG-Pype, a comprehensive yet easy-to-use pipeline built on open-source MNE-Python functions. Grounded in the idea that rigorous manual data selection contributes to the quality of resting-state analyses, our software guides users through a repro-ducible workflow focused on manual epoch and bad-channel selection. While this manual review is central, Independent Component Analysis (ICA) can be applied for additional data cleaning. qEEG measures can be calculated with a separate module after preprocessing. By utilizing a standardized workflow in an open-source application, EEG-Pype promotes reproducibility and transparency. In this paper, we explain the choices that were made in constructing EEG-Pype as a general resting-state EEG pipeline and demonstrate its application in a step-by-step manner.

## Design and implementation

### Workflow philosophy

Resting-state EEG, recorded in the absence of specific tasks, is valuable for the study of intrinsic brain activity [5]. Intrinsic activity is crucial for understanding brain function, as it accounts for the vast majority of the brain's energy consumption and plays a vital role in how the brain interprets, responds to, and predicts environmental demands [6]. Resting-state activity and interactions in resting-state networks [7] can be characterized using metrics based on spectral, functional connectivity, and network analysis. Although a one-size-fits-all approach in EEG analysis does not exist, preprocessing of resting-state EEG relies on fairly standard steps, enabling construction of a general preprocessing pipeline that allows researchers to perform EEG preprocessing easily and in a reproducible manner.

Various preprocessing strategies are possible, each presenting certain trade-offs. Automated methods, such as ICA, are powerful for identifying and removing artifacts from continuous recordings and can reduce preprocessing subjectiv-ity to some extent [8,9]. The effectiveness of automated techniques can be compromised by non-stationary signals or complex, transient artifacts. Sole reliance on global data cleaning techniques carries the risk of either incomplete artifact removal or the distortion of neural signals if artifactual components are not clearly separable. Visual inspection, though more labor-intensive, offers a high degree of precision in identifying a broader range of artifacts, especially those that are subtle or atypical. Therefore, the main focus in EEG-Pype lies on increasing the efficiency of manual data selection for resting-state EEG.

### Workflow for preprocessing

EEG-Pype guides users through a sequential preprocessing workflow via its graphical user interface (GUI). The steps correspond to functions within the provided Python script, which wrap MNE-Python functionalities. The core functionality is defined in the main script, while a separate settings script allows more advanced users to change default settings. The different steps making up the EEG-Pype preprocessing and quantitative analysis pipeline are detailed in Table 1.

Analysis with EEG-Pype uses a batch-setup, allowing for the import of any number of equivalent raw EEG files. By using MNE-Python's generic I/O interface, EEG-Pype supports the automatic detection and import of a broad range of standard electrophysiological formats (e.g., BDF, EDF, FIF, BrainVision, CNT, EEGLAB, EGI). Additionally, a custom parser is included to handle plain ASCII (*.txt) files with automatic delimiter detection and configurable header parsing. Once raw EEG files are selected, several choices regarding optional signal processing and data inspection steps can be made, which are accompanied by "More info" buttons linking to relevant online information for additional clarification. Next, EEG-Pype iterates through the selected EEG files one at a time. EEG-Pype applies an electrode montage that couples spatial coordinates to electrode positions based on channel name. There is an option to change channel names to match those included in the MNE-Python standard montage. Next, the user has the option to completely drop channels from the current file. This function is normally used for empty or non-physiological channels.

**Table 1. Overview of the various steps comprising the EEG-Pype preprocessing and quantitative analysis workflow. AECc: corrected Amplitude Envelope Correlation, JPE: Joint Permutation Entropy, PLI: Phase Lag Index.**

| Pipeline stage | Core function | Key features and options |
| --- | --- | --- |
| **1. Data Input** | Load raw EEG files | • Batch processing of multiple files<br>• Supported formats: BDF, EDF, FIF, BrainVision, CNT, ASCII (.txt) and other MNE-supported formats |
| **2. Preprocessing** | Clean and prepare data | Visual Inspection:<br>• View raw Power Spectral Density<br>• Scroll through continuous data<br>Artifact & Channel Handling:<br>• Drop non-physiological channels<br>• Manual rejection of bad channels (marked for interpolation)<br>• Independent Component Analysis for artifact removal<br>• Spherical spline interpolation<br>Filtering & Referencing:<br>• Broadband filtering<br>• Optional average referencing |
| **3. Epoching** | Select clean data segments | • Segment continuous data into epochs of user-defined length<br>• Manual visual rejection of bad epochs |
| **4. Source Analysis** | Reconstruct brain source activity | • Optional LCMV beamforming<br>• Uses fsaverage template MRI and Desikan-Killiany atlas |
| **5. Data Export** | Save processed data and logs | • Export clean epochs or continuous signal<br>• Filter into canonical frequency bands (delta, theta, alpha, beta) upon export<br>• Create log files and reproducible batch-processing files (.pkl) |
| **6. Quantitative Analysis** | Calculate qEEG metrics from processed data | Spectral Analysis: Absolute/relative power, peak frequency<br>Functional Connectivity: PLI, AECc, JPE<br>Network Analysis: Minimum Spanning Tree<br>Complexity: Permutation, Sample, and Approximate Entropy |

Users can then inspect an unfiltered power spectrum, including MNE-Python functionality to inspect spectra of specific channels to identify non-physiological channels. After this, users can scroll through the EEG and select channels with excessive noise or flat channels that will later be interpolated. Signal amplitude and zoom level can be altered using MNE interactive plots' standard controls. Before interpolating channels, the signal is broadband filtered to remove slow drift and high frequency noise. Independent component analysis (ICA) can be performed using the standard MNE-Python implementation [3]. This functionality is centered around the perform_ica() function in the EEG-Pype source code. ICA calculation is based on a bandpass filtered (1.0-47.0 Hz) copy of the EEG, excluding channels selected for interpolation in its calculation. This filtering step improves the subsequent decomposition, as ICA is more effective at separating sources when slow drifts and high-frequency noise, which can otherwise dominate the signal variance, are removed [10]. Excluding channels marked for interpolation during ICA calculation is also a critical step since their inclusion would violate the statistical assumption of source independence and could corrupt the decomposition [10]. A custom number of total ICA components can be chosen, influencing the number of components the signal is decomposed to (default: 25 components). To aid in selecting possibly artifactual ICA components, three plots are shown: head plots showing the spatial distribution

of the components, temporal plots showing the time series of each component across the entire measurement (Fig 2) and optionally a list of components with an estimated source (i.e., brain, eye, muscle, heart, line noise, channel noise or other) and confidence percentages based on MNE-Python's ICALabel functionality.

Selected channels are then interpolated using spherical spline interpolation. A second power spectrum depicting the bandpass filtered EEG is shown. Optionally, a scalar linearly constrained minimum variance (LCMV) beamformer is applied to an average-referenced copy of the signal. The beamformer spatial filter is constructed using the Freesurfer Average (fsaverage) template MRI and a corresponding boundary element model to describe the volume conductor, with an equivalent current dipole assumed as the source model. In our current implementation, the source space is defined as a grid of points derived from the Desikan-Killiany atlas [11], containing 68 cortical regions. In our experience, this atlas is suitable for 64-channel EEG, although higher density atlases may be used for EEG with higher measurement density. The orientation of each dipole is optimized to maximize output power. Beamformer weights are calculated using the data covariance matrix derived from the broadband-filtered EEG signal and a noise covariance matrix. The noise covariance is estimated as a diagonal matrix containing the variance measured at each sensor (i.e., the diagonal of the data covariance matrix). To improve the stability of the covariance matrix inversion, Tikhonov regularization is applied ($\lambda = 0.05$). This adds 5% of the mean sensor variance to the diagonal of the covariance matrix, ensuring numerical stability while preserving spatial resolution [3]. Finally, unit-noise-gain weight normalization is used to prevent localization bias. The current beamformer approach allows for simplified, easy access to source reconstruction, without the need for individual anatomical MRIs. This approach does not currently allow for the easy use of custom MRIs.

After an average reference is optionally applied to the EEG, the user is shown an MNE-Python epoch selection window utilizing a user-set epoch length (Fig 3). Here, the EEG is shown in sensor space. In the epoch selection window, the standard MNE-Python navigation options are present. Custom vertical markers denote whole seconds. By clicking on the signal in the current view, epochs can easily be deselected from the current selection of high-quality epochs, allowing for quick manual epoch selection in a matter of minutes. Data can be filtered in adjustable (canonical) frequency bands before the final signal export in ascii (*.txt) format. This approach reduces edge artifacts by filtering on the continuous signal. Additionally, custom down sampling can be applied, reducing exported file size. EEG-Pype can also be run without epoch selection, in which case the continuous EEG signal is saved in ascii (*.txt) format.

In addition to the processed files, log and batch-processing files are created. The log file contains detailed information of each setting and step taken, allowing for careful retracing of data analysis steps. The batch processing file (*.pkl file) can be loaded in case the user wishes to rerun a batch at a later point in time, automatically loading the previous channel and epoch selections.

## Quantitative analysis

EEG-Pype can be used to calculate several quantitative EEG measures after preprocessed EEG data is saved. Power Spectral Density (PSD) measures are calculated on the epochs saved in the broadband, while entropy, functional connectivity and by extension topology measures are calculated on all previously exported frequency bands. This functionality utilizes parallel processing capabilities to increase throughput. A separate GUI allows for easy selection of EEG measures and data. The following section describes which measures are included.

Power Spectral Density (PSD) can be computed using either Multitaper, Welch, or Fast Fourier Transform methods. Absolute and relative power within canonical frequency bands can be calculated based on the computed PSD. Frequency band cut-offs can easily be changed to suit specific analyses. Additionally, peak frequency within a user-defined range can be calculated from the broadband PSD. Peak frequency is calculated using a peak detection algorithm from the Signal Python package to increase precision [12]. Spectral variability, quantified as the coefficient of variation of relative band power over time, can also be computed using a sliding window approach on concatenated broadband epochs. Window

lengths for Welch PSD and spectral variability, which impact frequency resolution, can easily be changed from their default values in the GUI.

Permutation entropy [13], sample entropy [14] and approximate entropy [15] can be calculated, with user-configurable parameters tau or time step (permutation entropy), embedding dimension (sample entropy and permutation entropy) and tolerance (approximate entropy).

Functional connectivity can be assessed using three measures with (optional or inherent) correction for the effects of volume conduction/field spread. The phase lag index (PLI) [16], amplitude envelope correlation (AEC) [17] and joint permutation entropy (JPE) [18,19] can be calculated. For AEC, orthogonalization [17] can be applied to mitigate spurious correlations due to volume conduction. For JPE, parameters tau or time step can be altered in the GUI.

Measures of network topology, based on a so-called minimum spanning tree (MST) approach [20] can be calculated for connectivity matrices derived from PLI or (corrected) AEC. This method involves constructing an MST from the connectivity matrix (inverted to use maximum connection strength and treated as a weighted graph) using the Kruskal algorithm and calculating various metrics that characterize network topology, providing insights into the network's organization and efficiency [20].

Results are aggregated across epochs for each subject and condition and saved into an Excel file. This file includes whole-brain (or source-region) averaged measures, the number of epochs averaged per condition, and a log of the analysis parameters used. Optionally, results averaged per channel/region-level instead of whole-brain, averaged connectivity matrices (PLI, (corrected) AEC) and/or the derived MST adjacency matrices can be saved for further investigation or visualization.

## Results

To demonstrate the functionality and output of the EEG-Pype workflow, we processed the sample 64-channel resting-state EEG measurement provided with the software (a BioSemi.BDF file: EEG-Pype_test_EEG.bdf, located in the "EEG-Pype/src" folder once EEG-Pype is installed). This section details the reproducible, step-by-step application of the pipeline, showcasing the key outputs at each stage from raw data loading to the generation of quantitative metrics. The complete preprocessing for the approximately ten-minute sample recording took around two minutes. On our Github page (https://github.com/yorbenlodema/EEG-Pype), a video walkthrough of this process is available.

### 1. Initial inspection and filtering

Upon loading the raw.BDF file, EEG-Pype first presents an unfiltered PSD plot. This initial visualization allows the user to assess the overall data quality and identify prominent artifacts, such as line noise. This plot can be saved using the built-in MNE-Python save functionality by pressing the save icon. For the sample data, a distinct peak at 50 Hz was visible, corresponding to power line interference (Fig 1A). Following this inspection, a broadband finite impulse response (FIR) filter (0.5-47 Hz) was automatically applied to the data. The effectiveness of this step is demonstrated by comparing the PSD before and after filtering, where the 50 Hz peak is fully removed (Fig 1B).

### 2. Artifact identification and removal using independent component analysis

Prior to running ICA, the user is prompted to identify and drop non-EEG or empty channels. For the sample data, nine such channels were dropped (channels S1-Status in the selection window), resulting in 64 analyzable channels. Next, the user manually selects channels with excessive non-physiological noise to be excluded from the ICA computation but retained for later interpolation. Based on the filtered data, no noisy channels were marked as bad.

EEG-Pype then computes the ICA decomposition. The resulting components are displayed using MNE-Python's interactive plots, showing both the time course, spatial topography and optionally an automatic labeling and of independent components based on MNE-ICALabel (Fig 2). In the sample data, two components with a clear frontal topography

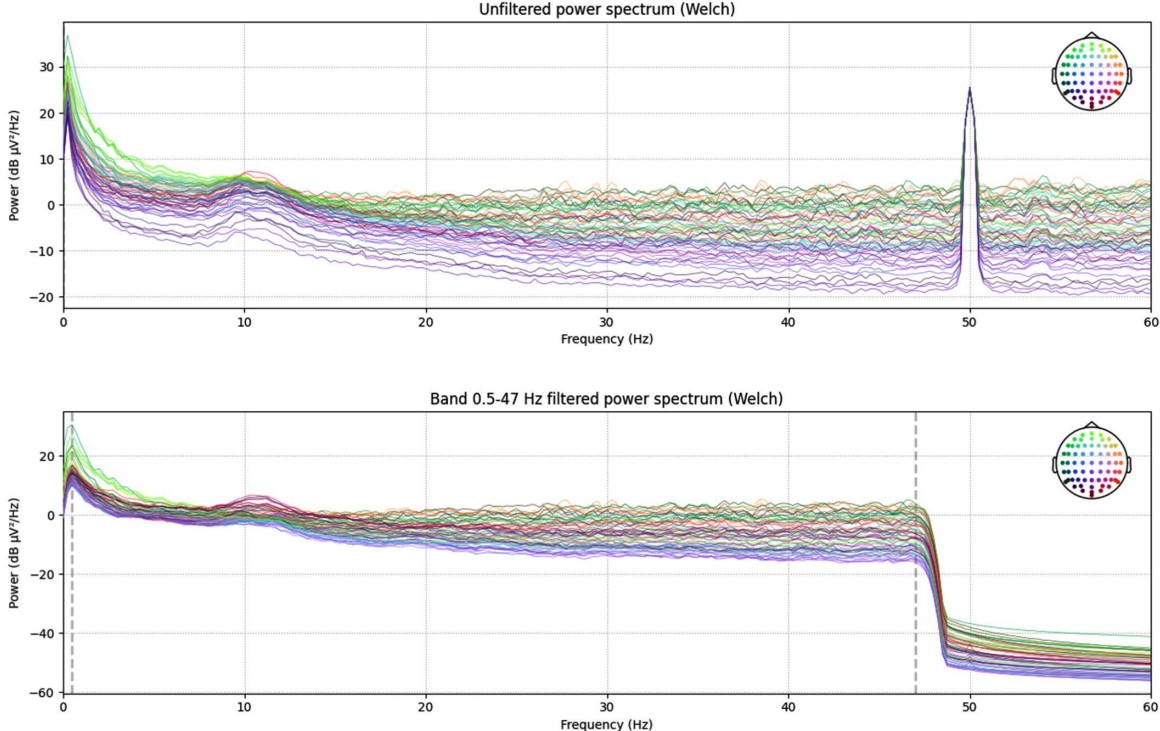

**Fig 1. Raw and Filtered Power Spectrum for the test EEG measurement. (A)** The power spectral density of the raw sample data, showing a prominent 50 Hz line noise artifact. **(B)** The power spectral density of the same data after the application of a 0.5-47 Hz band-pass filter, demonstrating the effective removal of line noise. The dotted lines correspond to the applied broadband filter cutoff values. dB: decibels, Hz: hertz, µV: microvolt.

and a time course characteristic of eye blinks were identified and selected for removal: ICA000 and ICA004. Note that ICA decomposition is stochastic, therefore providing slightly different results each time. Since muscle artefacts only affect parts of the measurement, the remaining components were retained, and the cleaned signal was reconstructed by projecting the noise components out of the data.

### 3. Bad channel interpolation and epoch rejection

Following ICA, the continuous, cleaned data was then segmented into eight-second epochs. EEG-Pype presents these epochs in an interactive window for manual rejection (Fig 3). This interface allows the user to scroll through the data and efficiently discard epochs containing transient artifacts such as muscle activity by simply clicking on them. For the sample data, from an initial total of 84 epochs, 29 were rejected, yielding 55 high-quality epochs for subsequent analysis.

### 4. Data export

Upon completion of the pipeline, EEG-Pype saved the processed data. For the sample file, this resulted in 330 files: 55 selected epochs saved for six frequency bands (broadband, delta, theta, alpha, beta1 and beta2).

### 5. Deriving quantitative EEG measures

The cleaned epoch files generated during preprocessing serve as the input for EEG-Pype's quantitative analysis module. This module uses a separate GUI to batch-process the data and compute a range of specified metrics (Fig 4).

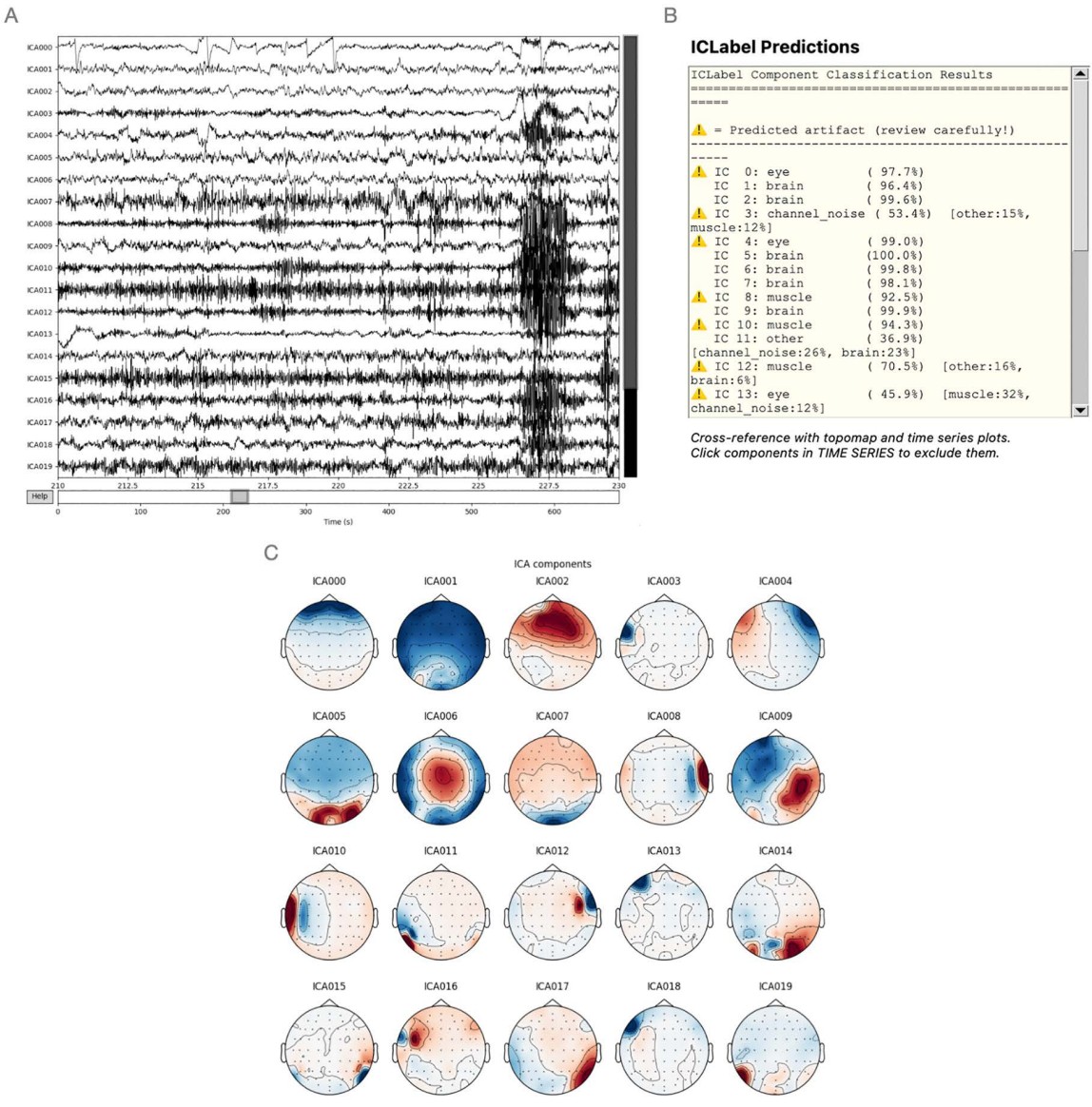

**Fig 2. Independent Component Analysis for the test EEG measurement. (A)** The time series plot shows the EEG signal over the entire measurement duration after ICA decomposition. The window shown here contains (eye) movement artifacts, with eye blinks clearly distinguishable in component ICA000 and muscle activity in multiple other components. The automatically generated IC labels and the algorithm confidence level in percentages, seen in **(B)**, can help to support ICA decision making. The topographical plot (C) confirms that component ICA000 corresponds to ocular artifacts, as can be seen from the field distribution with a maximum over frontal regions. ICA: Independent Component Analysis, s: seconds.

Using the 55 clean, pre-filtered epochs from the sample data, we computed several quantitative measures. Power spectral density was calculated using the Multitaper method. From this, we derived absolute and relative power in canonical frequency bands, as well as the peak frequency within the theta-alpha bands (4–13 Hz). We also calculated permutation entropy, sample entropy and approximate entropy as measures of signal complexity. Joint permutation entropy, phase lag index and corrected amplitude envelope correlation were calculated as measures of functional connectivity. Minimum spanning tree metrics were calculated on phase lag index and corrected amplitude envelope correlation connectivity matrices. Example whole-brain averaged results are shown in Table 2.

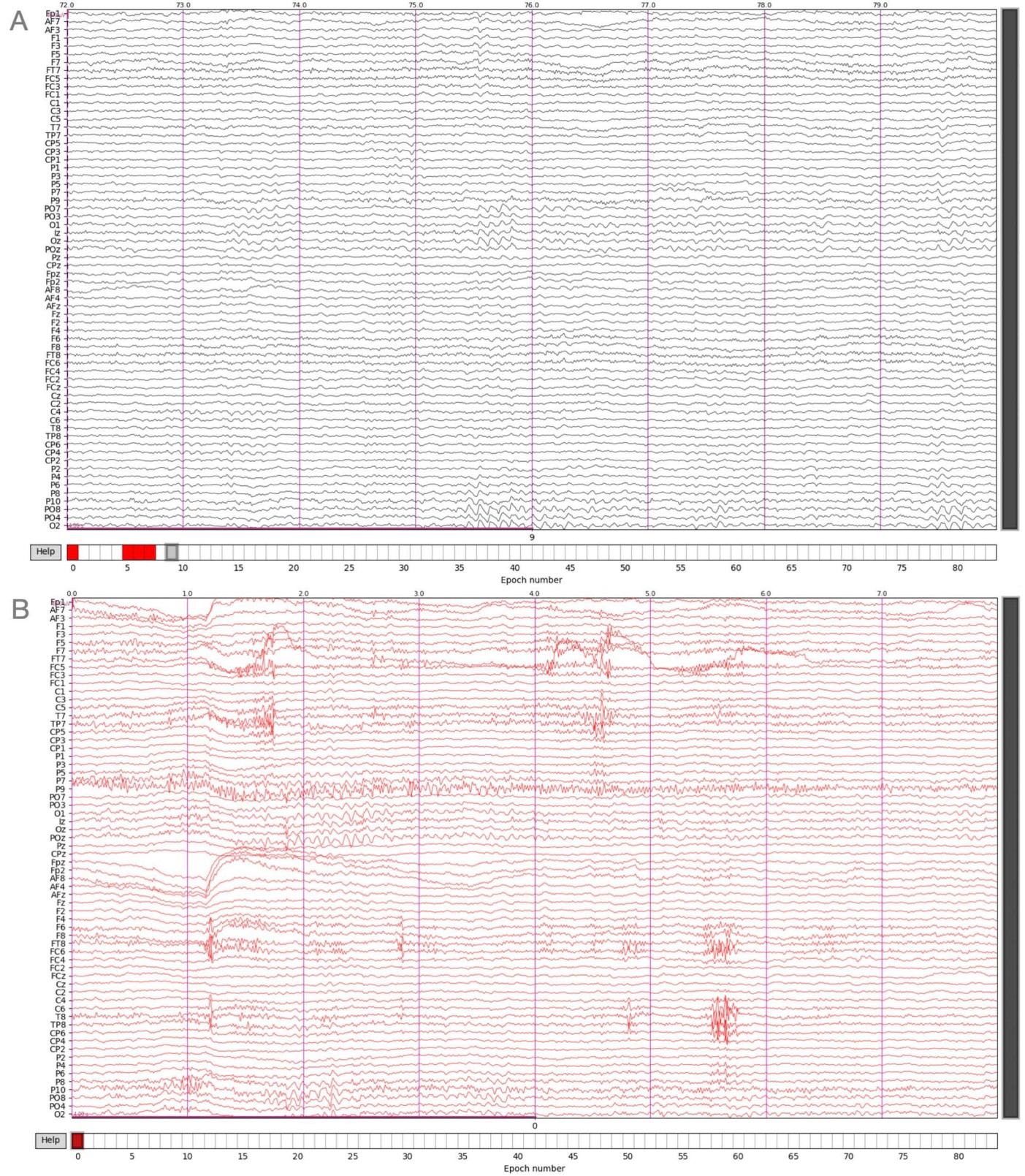

**Fig 3. MNE-Python interactive epoch selection windows. (A)** An example of an eight-second epoch with adequate quality. Magenta vertical markers denote whole seconds. **(B)** An example of an epoch with more artifacts present. The red color means the epoch will be discarded.

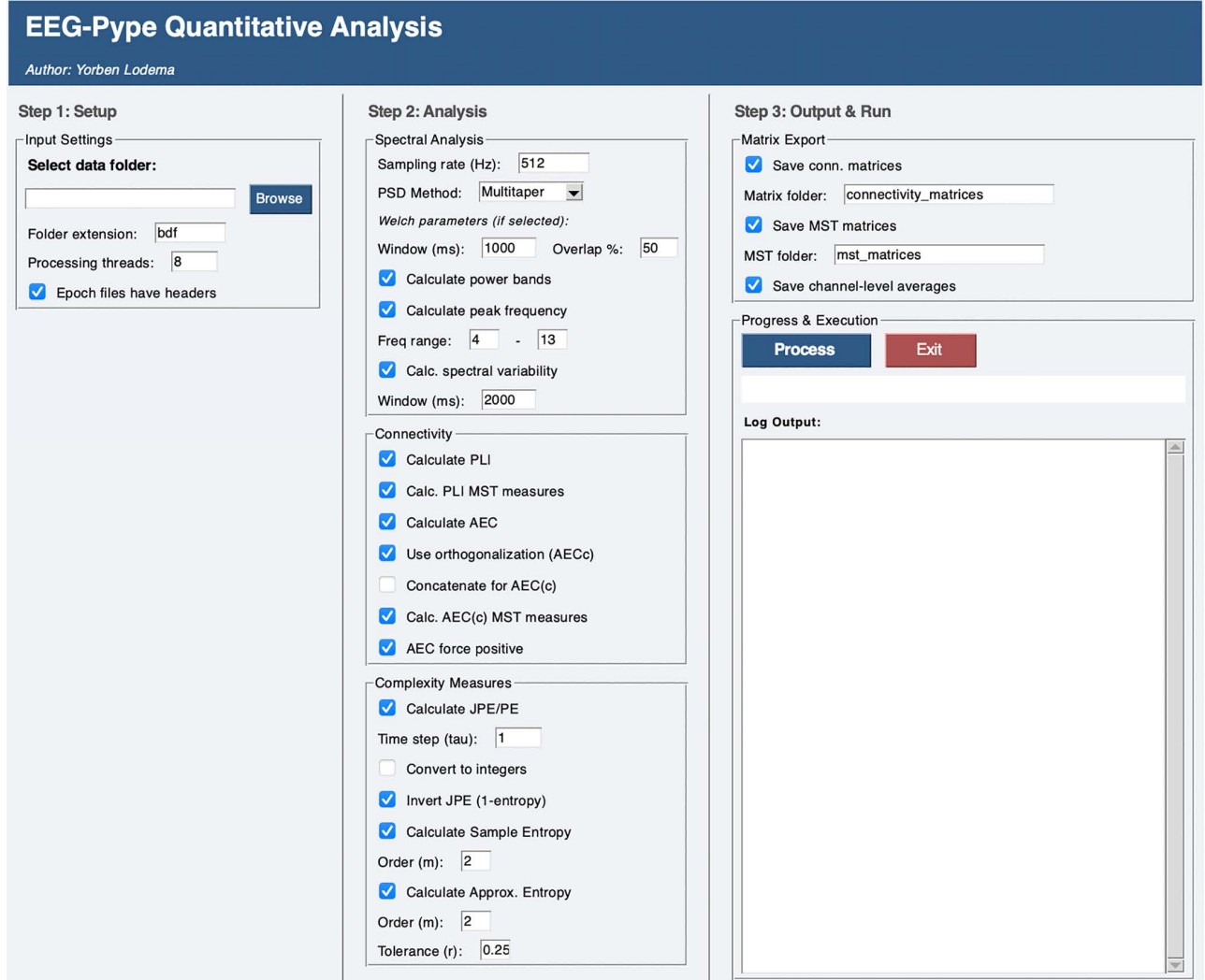

**Fig 4. The graphical user interface used for the quantitative analysis module of EEG-Pype.** Here, it can be seen that all possible quantitative metrics were selected. In addition, the optional output of connectivity matrices, MST matrices and channel-level averages (metric values per channel or brain area) was selected.

## Discussion

EEG-Pype was developed to facilitate access to the EEG preprocessing methods available through MNE-Python and easy quantitative EEG analysis. By providing a GUI-driven workflow, EEG-Pype aims to lower the barrier to resting-state EEG analysis, especially for clinicians or students. In addition, the use of a standard pipeline promotes consistency and reproducibility by reducing methodological variability. We believe our approach provides a shallower learning curve for users interested in a standard resting-state workflow. In addition, EEG-Pype can promote FAIR use of research data through logging processing steps and allowing for reproducible analyses.

Compared to other MATLAB-based alternatives like FieldTrip [21], EEGLAB [1] and Brainstorm [2], EEG-Pype is more focused on ease of use while still offering a fully functional pipeline from raw resting-state EEG to readily analyzable preprocessed data and often used quantitative spectral, complexity, functional connectivity and network

**Table 2. For the metrics calculated on the test EEG file (except minimum spanning tree metrics), the whole brain, epoch-averaged values are shown. Where applicable, these results are limited to theta and alpha bands, while the actual software output contains many more bands and non-epoch-averaged values. EEG: electroencephalography, Hz: Hertz, PSD: power spectral density.**

| Epoch used | Category | EEG metric | Value |
| --- | --- | --- | --- |
| Broadband | PSD | Peak frequency | 9.18 Hz |
| Broadband | PSD | Theta relative power | 0.224 |
| Broadband | PSD | Alpha relative power | 0.170 |
| Broadband | PSD | Theta spectral variability | 0.513 |
| Broadband | PSD | Alpha spectral variability | 0.545 |
| 4.0-8.0 Hz | Complexity | Theta permutation entropy | 0.304 |
| 4.0-8.0 Hz | Complexity | Theta sample entropy | 0.280 |
| 4.0-8.0 Hz | Complexity | Theta approximate entropy | 0.234 |
| 8.0-12.0 Hz | Complexity | Alpha permutation entropy | 0.348 |
| 8.0-12.0 Hz | Complexity | Alpha sample entropy | 0.440 |
| 8.0-12.0 Hz | Complexity | Alpha approximate entropy | 0.435 |
| 4.0-8.0 Hz | Functional connectivity | Theta joint permutation entropy | 0.437 |
| 4.0-8.0 Hz | Functional connectivity | Theta phase lag index | 0.116 |
| 4.0-8.0 Hz | Functional connectivity | Theta corrected amplitude envelope correlation | 0.060 |
| 8.0-12.0 Hz | Functional connectivity | Alpha joint permutation entropy | 0.423 |
| 8.0-12.0 Hz | Functional connectivity | Alpha phase lag index | 0.116 |
| 8.0-12.0 Hz | Functional connectivity | Alpha corrected amplitude envelope correlation | 0.116 |

topology measures. Two other available MNE-Python GUI-based EEG preprocessing packages, Meggie [22] and MNELAB [23], share our goal of providing an accessible way to utilize MNE-Python functions for a non-coding audience. Our implementation differs from these two applications in providing a complete EEG preprocessing pipeline without the need for further software. The central role of manual data selection after segmenting data into epochs in our software is another distinctive feature. Finally, EEG-Pype differs from Meggie and MNELAB in focusing on resting-state EEG analysis. Beyond these contemporary MNE-based GUIs, EEG-Pype also draws conceptual inspiration from BrainWave [4], an earlier EEG (pre)processing program that similarly aimed to make complex EEG analysis accessible to clinicians and researchers. Our emphasis on a guided workflow with manual epoch selection is philosophically aligned with the approach used by BrainWave. By building on MNE-Python rather than a compiled Java environment, we achieve greater interoperability with the wider ecosystem of data science and machine learning tools and allow for community participation in development.

A strength of EEG-Pype is that it can improve efficiency and reproducibility of EEG preprocessing and analysis, while also lowering the amount of knowledge needed to perform these analyses. Analysis can be sped up compared to other software solutions due to parallel processing and the option to re-run previous analyses. Additionally, its open-source nature allows for external contributions and transparent ongoing development. A limitation is that our pipeline is semi-rigid, allowing for less customization than other EEG preprocessing solutions, though this choice can be seen as a trade-off between ease-of-use and customizability. Another limitation concerns the current beamforming approach, where the used noise covariance estimation and template-based beamforming represent simplifications. These were made for reasons of accessibility and ease of implementation, but reduce precision compared to best practices that utilize individual MRI data and dedicated noise recordings.

In conclusion, EEG-Pype offers an easy-access, open-source solution for a complete resting-state EEG analysis pipeline with extensive capabilities for preprocessing and quantitative analyses.

## Availability and future directions

On our Github repository (https://github.com/yorbenlodema/EEG-Pype) instructions for installation and use of EEG-Pype can be found, together with a video tutorial explaining the use of the software. The toolbox is available under an Apache-2.0 license.

To allow users to test the software and familiarize themselves with its workflow, we provide a sample resting-state EEG measurement in the GitHub repository. The test data is a 64-channel resting-state measurement in the BioSemi Data Format (.BDF) and is suitable for demonstrating the complete functionality of EEG-Pype. Users can follow the workflow detailed in the Results section to familiarize themselves with the software.

External contributions to EEG-Pype can be made via our Github repository and are welcomed. Development is expected to continue for at least several years, keeping up with the development of MNE-Python. We see possibilities to add more sophisticated source reconstruction (i.e., using minimum-norm estimate and/or custom templates) and to expand the quantitative features that can be calculated with the quantitative analysis module.

## Supporting information

**S1 File. This archive contains a full version of EEG-Pype at the moment of publication.** This archive serves as a reference of the functionality of EEG-Pype at this point in time, which may be expanded in the future. This archive does not include our test.bdf EEG file due to file size constraints, which is, and will stay, available on our GitHub page: https://github.com/yorbenlodema/EEG-Pype.
(ZIP)

## Acknowledgments

We would like to thank Arno Fennema for contributing to the functional connectivity analysis which later became part of the quantitative analysis module of EEG-Pype. We would also like to thank Professor C.J. Stam for his work on making EEG (pre)processing more accessible with BrainWave, serving as an inspiration for the current software.

## Author contributions

**Conceptualization:** D. Yorben Lodema, Willem de Haan, Edwin van Dellen.

**Data curation:** D. Yorben Lodema.

**Formal analysis:** D. Yorben Lodema, Herman J. van Dellen.

**Methodology:** D. Yorben Lodema, Herman J. van Dellen, Margot van Hest, Arjan Hillebrand.

**Software:** D. Yorben Lodema, Herman J. van Dellen, Willem de Haan, Margot van Hest, Arjan Hillebrand, Edwin van Dellen.

**Supervision:** Arjan Hillebrand, Edwin van Dellen.

**Visualization:** D. Yorben Lodema.

**Writing – original draft:** D. Yorben Lodema.

**Writing – review & editing:** Herman J. van Dellen, Willem de Haan, Margot van Hest, Arjan Hillebrand, Edwin van Dellen.

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
