## [Decision Letter · Decision Letter 0]

22 Jan 2026

PCOMPBIOL-D-25-02246

EEG-Pype: an accessible MNE-Python pipeline with graphical user interface for preprocessing and analysis of resting-state electroencephalography data

PLOS Computational Biology

Dear Dr. Lodema,

Thank you for submitting your manuscript to PLOS Computational Biology. After careful consideration, we feel that it has merit but does not fully meet PLOS Computational Biology's publication criteria as it currently stands. Therefore, we invite you to submit a revised version of the manuscript that addresses the points raised during the review process.

We look forward to receiving your revised manuscript.

Kind regards,

Daniel Bush

Section Editor

PLOS Computational Biology

Thomas Serre

Section Editor

PLOS Computational Biology

**Additional Editor Comments:**

The authors should ensure that the procedure for installing the GUI is made as straightforward as possible, and that the other support documentation is clear and error free, to maximise the potential impact of this software.

**Journal Requirements:**

2) Please ensure that the Title in your manuscript file and the Title provided in your online submission form are the same.

**Reviewers' comments:**

Reviewer's Responses to Questions

**Comments to the Authors:**

Reviewer #1: The paper describes an open source graphical user interface application “EEG-Pype” that uses MNE-Python functions. The application designed for smooth defined processing of resting state EEG data to use for researchers without programming experience.

The work demonstrates a functioning pipeline; however, it does not appear to introduce novel elements beyond what is already available in the literature. Its current rigidity (only resting state data) and the omission of basic functionality (e.g., triggers) constrain its practical utility. The tool may nonetheless provide value in narrow contexts. Intuitive graphical user interface removes some barriers for researchers to start processing their data, which is great for the community. And as the GitHub page with application is public the interface has a potential to grow if community needs more.

Major concerns:

1) My main concern is the installation instructions. Since “EEG-Pype” is designed for researchers without much technical experience, the current guide doesn’t feel very user-friendly for its target audience. I found myself struggling with the installation process as well.

Would it be possible to add another video demonstrating the full installation workflow, or include additional screenshots for each step? I think this would make the process much clearer for new users

2) The instructions on GitHub for the scrips have couple of inaccuracies:

a) in the section “Quantitative analysis module (separate script)”

the script is not in the folder “/EEG-Pype”, but in the folder “EEG-Pype/scr”

b) I would add where the test data in the GitHub. In the paper L238 mentions “a BioSemi.bdf file” as a tested data. However, the data name at the GitHub “EEG-Pype_test_EEG.bdf”

Minor concerns:

1) L151 – What is the default number of ICA components?

2) L170 Please explain the choice of regularization parameter?

3) Is it possible to save the figures? Not just data?

4) L243 please add the link to the GitHub page.

Suggestion:

Being able to extract the raw Python mne-python script generated through the interface could be very beneficial—both for collaboration and for making custom modifications to the processing pipeline.

Reviewer #2: The authors developed an interactive GUI that wraps up MNE-Python functions to provide: a) preprocessing and b) analysis pipelines for rsEEG signals. Overall, the authors aim to simplify access to standardized methods for rsEEG processing among researchers without programming experience. The latter could be particularly helpful in paving the way for clinicians involved in neuroscience research. This reviewer salutes the initiative and considers the manuscript worthy of publication with minor changes. Similarly, minor changes are suggested in the pipeline implementation, accounting for user experience and potential needs for research applications.

Minor recommendations for the manuscript:

• On page 9 (lines 126 & 127), please consider pointing the reader to the specific function and .py script for expanding the read raw functions.

• On page 9 (lines 143 & 144), please state in the manuscript the selected frequencies for the bandpass filter prior to ICA (1 – 47 Hz).

• On page 9 (lines 143 & 144), please direct the readers to the perform_ica() function, as some users would like to modify ICA parameters or method beyond the number of components.

• On page 12 (lines 210 & 211), please state that entropy-based features are not calculated across the channel time series (broadband) but over a filtered version for each frequency band (returning the entropy by band based on previously reported methods by the authors https://pmc.ncbi.nlm.nih.gov/articles/PMC12586465/). Although this clarification can be inferred from the tables, declaring it will provide a clearer methodology for the reader.

Recommendations for the GUI & Pipeline implementation:

A) Preprocessing module – Set montage & Read raw files

• In the “Enter filetype” window, consider adding a “More Info” button leading to the MNE “work with sensor data” tutorial, or ideally, consider adding some images generated with the plot_sensors() so the user can “preview” the expected ch_names and inspect if the montage is appropriate for the data.

• Also consider expanding (or pointing the author to the set_montage function) to the 10-10 and 10-05 montages, which can be suitable for datasets with a high density channel number.

• It would be great to set mne.io.read_raw() as the default reading function (as it wraps over much more file formats than the currently available). https://mne.tools/stable/generated/mne.io.read_raw.html

• Following the above comment, if the authors want to keep the individual reader functions, please state in the manuscript that “.cnt” supported files come from NeuroScan amplifiers, not ANT Neuro (which require read_raw_ant()). https://mne.tools/stable/auto_tutorials/io/20_reading_eeg_data.html

B) Preprocessing module – Set Ch_names

• If ch_names doesn’t conform to the expected names (due to suffixes like “-AVG” or prefixes like “EEG”), it would be nice to add a destrip function to run over all channels with suffixes or prefixes. More importantly, authors can consider applying the standardize() function (it changes the midline “Z” suffix to lowercase, the latter is required for some devices like NeuroScan), see https://mne.tools/stable/generated/mne.datasets.eegbci.standardize.html

• It may also be useful to support inheriting the “corrected” ch_names from one EEG file to subsequent files, so the user doesn’t have to repeat the correction for every file when channel names are consistent across recordings.

C) Preprocessing module – Set montage, ICA & Filters

• Consider allowing the user to reassign ch_names if mistakes/typos happen. In my use case, I edited “F8-AVG” and wrote “F7” then it was not possible to reassign it or repair the mistake.

• Similarly, it would be advisable to assert that there are no duplicate ch_names before running the preprocessing pipeline for a given file.

• For non-experienced researchers assessing spectral and topographical patterns of brain-like and non-brain-like Independent Components, it would be advantageous to incorporate MNE ICALabel (https://mne.tools/mne-icalabel/stable/index.html) into the pipeline, returning the predicted label for each IC, as an optional tick check, so the user can also get assistance in the IC selection.

D) Analysis module

• From the user perspective, I would recommend rearranging the spatial position of each box (Input Settings, Matrix Export, Complexity Measures…) so that the qEEG analysis GUI resembles each of the steps of the pipeline.

• Also, consider if JPE should be placed in the Connectivity box.

**Have the authors made all data and (if applicable) computational code underlying the findings in their manuscript fully available?**

Reviewer #1: Yes

Reviewer #2: Yes

PLOS authors have the option to publish the peer review history of their article (what does this mean? ). If published, this will include your full peer review and any attached files.

**Do you want your identity to be public for this peer review?** For information about this choice, including consent withdrawal, please see our Privacy Policy .

Reviewer #1: No

Reviewer #2: **Yes:** Alberto Jaramillo-Jimenez

**Figure resubmission:**

**Reproducibility:**



---

## [Decision Letter · Decision Letter 1]

19 Feb 2026

Dear Mr. Lodema,

We are pleased to inform you that your manuscript 'EEG-Pype: an accessible MNE-Python pipeline with graphical user interface for preprocessing and analysis of resting-state electroencephalography data' has been provisionally accepted for publication in PLOS Computational Biology.

Best regards,

Daniel Bush

Section Editor

PLOS Computational Biology

Thomas Serre

Section Editor

PLOS Computational Biology

Reviewer's Responses to Questions

**Comments to the Authors:**

Reviewer #1: I thank the authors for their thoughtful revisions and for addressing all of my previous comments clearly and comprehensively. I have no further comments to add.

Reviewer #2: I appreciate the considerable effort the authors have invested in addressing all of my previous comments on the manuscript and software implementation. The improvements made to EEG-Pype reflect careful attention to both the developer and end-user perspectives. As a clinician with a background on rsEEG research (and a strong curiosity for coding), I greatly value their efforts to bridge the gap between technical aspects of signal processing and practical clinical usability (e.g., extracting reliable and reproducible features).

Overall, the revisions have significantly strengthened the manuscript and the presented framework, so I am pleased to recommend this paper for publication.

**Have the authors made all data and (if applicable) computational code underlying the findings in their manuscript fully available?**

Reviewer #1: Yes

Reviewer #2: Yes

PLOS authors have the option to publish the peer review history of their article (what does this mean? ). If published, this will include your full peer review and any attached files.

**Do you want your identity to be public for this peer review?** For information about this choice, including consent withdrawal, please see our Privacy Policy .

Reviewer #1: No

Reviewer #2: **Yes:** Alberto Jaramillo-Jimenez

---

## [Editor Report · Acceptance letter]

PCOMPBIOL-D-25-02246R1

EEG-Pype: an accessible MNE-Python pipeline with graphical user interface for preprocessing and analysis of resting-state electroencephalography data

Dear Dr Lodema,

I am pleased to inform you that your manuscript has been formally accepted for publication in PLOS Computational Biology. Your manuscript is now with our production department and you will be notified of the publication date in due course.

With kind regards,

Anita Estes
